# Monoclonal Antibody Engineering and Design to Modulate FcRn Activities: A Comprehensive Review

**DOI:** 10.3390/ijms23179604

**Published:** 2022-08-24

**Authors:** Yanis Ramdani, Juliette Lamamy, Hervé Watier, Valérie Gouilleux-Gruart

**Affiliations:** 1Service de Médecine Interne Immunologie Clinique, CHU de Tours, F-37032 Tours, France; 2EA7501, GICC, Faculté de Médecine, Université de Tours, F-37032 Tours, France; 3Laboratoire D’immunologie, CHU de Tours, F-37032 Tours, France

**Keywords:** FcRn, monoclonal antibody, antibody engineering, recycling, transcytosis

## Abstract

Understanding the biological mechanisms underlying the pH-dependent nature of FcRn binding, as well as the various factors influencing the affinity to FcRn, was concurrent with the arrival of the first recombinant IgG monoclonal antibodies (mAbs) and IgG Fc-fusion proteins in clinical practice. IgG Fc–FcRn became a central subject of interest for the development of these drugs for the comfort of patients and good clinical responses. In this review, we describe (i) mAb mutations close to and outside the FcRn binding site, increasing the affinity for FcRn at acidic pH and leading to enhanced mAb half-life and biodistribution, and (ii) mAb mutations increasing the affinity for FcRn at acidic and neutral pH, blocking FcRn binding and resulting, in vivo, in endogenous IgG degradation. Mutations modifying FcRn binding are discussed in association with pH-dependent modulation of antigen binding and (iii) anti-FcRn mAbs, two of the latest innovations in anti-FcRn mAbs leading to endogenous IgG depletion. We discuss the pharmacological effects, the biological consequences, and advantages of targeting IgG–FcRn interactions and their application in human therapeutics.

## 1. Introduction

FcRn is a very unique IgG Fc receptor encoded by the *FCGRT* gene, present on chromosome 19 in humans [1]. Theorized by Brambell et al. as controlling circulating IgG concentrations as early as 1964 [2], it was named neonatal Fc receptor for its capacity to transfer IgG from mother to fetuses and/or newborns (depending on the species) [3,4,5]. It was eventually classified as a member of the MHC class-I family after its crystal structure was resolved by Burmeister et al. [2,6]. FcRn clearly differs from the Fc gamma receptor (FcγR) family: its binding site on IgG Fc is located at distance from that of FcγR, its binding does not depend on IgG Fc glycosylation and it is not needed for IgG effector functions [6,7]. It is expressed in different cell types, such as endothelial and epithelial cells, and is also co-expressed with FcγR family members in hematopoietic cells of the myeloid lineage, such as neutrophils, monocytes/macrophages and dendritic cells [8,9]. After passive pinocytosis of the fluid phase, IgG binds FcRn in the endosomes at acidic pH (pH~6), escaping from catabolism and being released in the extracellular medium at near neutral pH 7.4, either on the apical side of endothelial cells (recycling) or on the opposite side of endothelial and epithelial cells (transcellular transport) [10,11]. The mechanism by which FcRn binds and recycles/transports albumin is similar to that of IgG, although occurring at a different stoichiometry (1:1 and 1:2 for albumin:FcRn and IgG:FcRn, respectively), and involving a different non-overlapping binding site on FcRn [12].

By a quirk of fate, the early structural and functional knowledge of FcRn was concurrent with the arrival of the first recombinant IgG monoclonal antibodies (mAbs) and IgG Fc-fusion proteins in clinical practice, both events having taken place in the late 1990s. IgG Fc–FcRn became a central subject of interest for the development of these drugs since a comfortable dosing regimen for patients and good clinical responses required a long plasmatic half-life and a good drug exposure, respectively, both of which depend on the IgG Fc–FcRn interaction. Consequently, the FcRn-related scientific literature exponentially increased as the number of IgG Fc-containing biologics grew. Understanding the routing of therapeutic IgG and that of immune complexes (ICs) also benefited from the recognition of the criticality of the pH dependency of the Fc–FcRn interaction [13,14]. Last but not least, manipulating this IgG Fc–FcRn interaction started to be a focus of interest either through Fc mutation or through FcRn blockade. These approaches have been translated into the clinic and are now renewing the field of prophylactic and long-term treatments with mAbs or IgG Fc-fusion proteins. In this review, we will retrace the different steps that have enabled the generation of these new therapeutic compounds.

## 2. Antibody Modifications That Enhance IgG Half-Life and Biodistribution

### 2.1. Mutations Close to the FcRn Binding Site, Increasing Affinity at Acidic pH

Monoclonal Ab engineering has benefited from many cellular and molecular studies and diverse technologies such as X-ray crystallography, random or site-directed mutagenesis, deuterium exchange mass spectrometry, surface plasmon resonance and bioinformatics to decipher the interactions and the affinity between IgG and FcRn [7,10,15,16,17,18,19,20]. The initial rationale in mAb development was that an increased affinity for FcRn while maintaining the pH dependency will enhance mAb half-life and therapeutic effects (spacing of doses, exposure to the drug). 

Attention was drawn to the binding zone between FcRn and IgG [6,21], which involves the CH2 and CH3 domains of IgG and is conserved among mammalian species [6,18,21,22]. At this interface, on the Fc side, histidines are the main residues responsible for ligand binding at acidic pH [10,23,24]. When these residues at positions H310 and H435 together with isoleucine 253 are mutated to alanine, IgG binding to FcRn is severely reduced (Eu numbering). The I253A, H310A and H435A combination of mutations (AAA) is quite useful in research to ascertain the participation of IgG binding to FcRn but has exhibited no therapeutic potential [25]. Conversely, many combined or isolated mutations affecting T250, M252, S254, T256, T307, E380, M428, H433 or N434 have been described to enhance mAb affinity to FcRn only at acidic pH~6 [26,27,28]. However, a good in vitro profile does not necessarily translate into a good in vivo profile. Indeed, few combinations of mutants eventually proved to have better PK properties, as exemplified by the T250Q/M428L (QL) mutation. This particular variant very well illustrates the species-specific differences and the difficulties in finding the right readout [29]. QL mutants have a 40-fold increased affinity to cynomolgus FcRn without increased half-life in the corresponding in vivo preclinical model [30], but the same QL mutation in a human IgG1 showed a 2.5-fold increased half-life in rhesus monkeys [31]. These results suggest that additional structural/physiological factors are influencing mAb half-life. However, some mutations or combinations of mutations increasing mAb half-life in humans are now clearly established, such as M428L/N434S (LS), M252Y/S254T/T256E (YTE), T250Q/M428L (QL) and T307A/E380A/N434A (AAA), as reviewed by Wang et al. and Kuo et al. [32,33]. The main mutations are summarized in Figure 1.

A specific increase in IgG binding at pH~6.5 is mandatory to ensure the release of IgG at neutral pH, as demonstrated by Yeung et al. and Deng et al. [29,34]. Indeed, these authors showed that IgG mutants with increased affinity to FcRn at both acidic and neutral pH exhibited similar or enhanced clearance as compared to their wild-type counterpart, highlighting the importance of the neutral-pH-dependent mechanism of IgG release in the extension of their half-life [29,34]. The relationship between enhanced mAb half-life and therapeutic effect was well described by Zalevski et al., who generated different IgG1 variants of bevacizumab (anti-VEGF mAb) and identified a lead with the LS (M428L/N434S) mutations presenting high pharmacokinetic performance associated with improved anti-tumor activity in mouse models [35].

The in vitro and in vivo validation of mutations extending mAb half-life led to the development of therapeutic antibodies in humans, mainly against viral agents such as respiratory syncytial virus: motavizumab, nirsevimab; human immunodeficiency virus-1 (broadly neutralizing HIV-1 antibodies (bNAbs)) or SARS-CoV-2: sotrovimab, tixagevimab + cilgavimab, amubarvimab + romlusevimab, but also in inflammatory diseases such as psoriasis, with an anti-IL-17A inhibitor: netakimab. These antibodies have LS (sotrovimab, bNAbs) [36,37], YTE (motavizumab, nirsevimab, tixagevimab + cilgavimab, amubarvimab + romlusevimab, netakimab) [38,39,40,41], or LA (adintrevimab) [42] mutations and are currently being clinically tested while other have already received market approval. The effects of the mutations on mAb pharmacokinetics are in accordance with those observed in animal studies. They lead to an increased half-life reaching 100 days for motavizumab [38], tixagevimab + cilgavimab [39] or adintrevimab [42], or five months with one injection for nirsevimab [43].

### 2.2. Modifications Outside FcRn Binding Site, Increasing Affinity for IgG Only at Acidic pH

The limitation of in silico approaches or directed mutagenesis in the FcRn binding site is therefore the unpredictability of its effect. Indeed, the combination of positive mutations does not always synergize, and sometimes the mutations even antagonize each other [44]. Martin et al. reported that mutations of amino acids located directly at the IgG–FcRn interface decreased the affinity in contrast to those located nearby, suggesting that minimum conformational changes could modify the affinity [16].

Experiments analyzing pharmacokinetic differences between IgG sharing the same Fab or Fc regions have highlighted other structural parameters affecting FcRn affinity and mAb half-life [45,46]. We and others have shown that IgG allotypes, modifications in the isoelectric point of complementarity-determining regions (CDRs) and light chain variations [47,48,49] could influence mAb half-life. For example, changes in CDR isoelectric point are able to increase FcRn affinity 79-fold [48]. This unpredictability has led to complementary approaches for developing mAbs with enhanced FcRn binding at acidic pH.

Using random mutagenesis combined with a pH-dependent phage display selection for FcRn binding, Monnet et al. have generated variants with increased FcRn binding at acidic pH [19]. Interestingly, they identified new mutations, located at distance from the FcRn binding site. Random mutations occurred all along the Fc region, some of them located in the hinge region (P228R or L and P230S) or in the upper part of the CH2 IgG domain (F241L, V264E and A330V). These mutations are located close to FcγRs and C1q binding regions on the Fc portion [50,51]. They can increase or decrease effector functions such as antibody-dependent cell cytotoxicity or complement-mediated cell cytotoxicity, depending on the mutations [52,53,54].

### 2.3. Mutations in the Vicinity of FcRn Binding Site Associated with pH-Dependent Modulation of Antigen Binding

The increase in mAb half-life has allowed, as discussed above, the maintenance of high plasma levels of therapeutic antibodies over longer periods of time, resulting in therapeutic benefits. However, the FcRn-dependent mAb-recycling is not interrupted when the mAb is attached to its antigen; i.e., FcRn recycles ICs as efficiently as free mAbs. This could be an issue, particularly in the case of autoantigens which are continuously produced and represent the main targets in human therapeutics. Firstly, this could result in plasma accumulation of the soluble antigen complexed with IgG [55], which could induce paradoxical pharmacological effects in case the mAb is not sufficiently neutralizing. Secondly, once bound to their antigen, the mAbs are no longer pharmacologically active and FcRn unnecessarily recycles them, possibly at the expense of free mAbs.

To solve this issue, antibodies capable of releasing their target in endosomes at acidic pH and then being recycled by FcRn freed from their antigen were generated. They were named “recycling” antibodies (recycling of a pharmacologically restored mAb). The repetition of these cycles enhances autoantigen clearance and contributes to improving free mAb pharmacokinetic parameters and to reducing therapeutic mAb dosage [56,57]. Igawa et al. provided the first proof of concept with tocilizumab, an anti-IL-6 receptor (IL-6-R) mAb. They engineered the tocilizumab CDR and named it PH2, conferring it a 3.8-fold lower affinity at acidic pH compared to neutral pH. Moreover, they associated an N434A mutation in the Fc region to further enhance its half-life through better binding to FcRn at acidic pH. The mutant tocilizumab PH2 gained a recycling ratio of 75% and a prolonged in vivo effect of 4 weeks reflecting improved pharmacodynamic parameters in this preclinical model [58].

This clever approach coupling the recycling effect to enhance antigen clearance and mutations in the Fc to obtain a higher half-life has been successfully pursued. Two anti-C5 mAbs derived from eculizumab (ravulizumab and crovalimab) with LS and M428L/N434A mutations, respectively [59,60], and a humanized anti-IL6R IgG2 mAb (satralizumab) with an N434A mutation have been successfully developed [29,61,62]. Ravulizumab and satralizumab have been approved in the USA and Europe for the treatment of paroxysmal nocturnal hemoglobinuria [63,64] and diseases of the optic neuromyelopathy spectrum, respectively [65,66]. Crovalimab is currently in phase I/II for the treatment of paroxysmal nocturnal hemoglobinuria [67,68].

To further enhance the efficiency of soluble antigen clearance, particularly for soluble antigens with high plasma concentrations, “sweeping” antibodies were generated. Like recycling mAbs, they can lose their antigen in the endosomes at acidic pH, but they displayed an enhanced affinity for FcRn at neutral pH, using mutations M252W/N434W, M252Y/N434Y, M252Y/N268E/N434Y (YEY) or M252Y/V308P/N434Y (YPY). This increased affinity for FcRn at neutral pH induces the persistence of IgG with FcRn complexes at the cell surface, ready to capture and internalize new antigens [69,70]. The team of Yang et al. showed that variants with a higher affinity for FcRn at neutral pH required a lower affinity for the antigen at acidic pH to maintain efficient clearance, contrarily to IgG variants while retaining their non-binding capacity to FcRn at neutral pH [71]. Moreover, increasing the affinity of an antibody for FcRn at neutral pH leads to its intracellular accumulation and elimination [28,72]. The main challenge of this technology is therefore finding a balance between removing as much antigen as possible and maintaining a sufficient level of antibody. Furthermore, it is interesting to note that despite the increased affinity for FcRn at neutral pH, these sweeping variants have no effect on endogenous IgG levels contrarily to the ABDEG developed by Argenx discussed later in Section 3, which has a much stronger affinity for FcRn at neutral pH [71]. The sweeping antibody technology has shown promising results in neutralizing the staphylococcal enterotoxin B in a murine model, decreasing the toxin levels, and opening a new way for treating some bacterial infections [73]. Finally, a recent proof of concept has been published by Bogen et al. who have generated a bispecific sweeping antibody targeting two tumor antigens, one CEACAM5, in a pH-dependent manner, and the other CEACAM6 in a pH-independent manner. CEACAM5 is a driving factor for metastases in colorectal cancer and has a soluble form and a membrane-bound form. The soluble form can trap antibodies targeting tumoral cells, diminishing their efficacy. This bispecific antibody recognizes the tumoral cell with its pH-independent arm and with its pH-dependent arm can target soluble and membrane-bound forms of CEACAM5, diminishing their levels [74].

## 3. Antibody Modifications Targeting the FcRn Recycling Pathway to Modulate Endogenous IgG Levels including IgG Autoantibodies

Most of the mutations listed in Section 2 have been created with the aim of increasing mAb half-life by enhancing the affinity to FcRn at acidic pH and thus reducing the dosing frequency. However, some mutations increasing FcRn binding at both acidic and neutral pH, have provided other therapeutic opportunities. Vaccaro et al. have developed the ABDEG technology, developed by Argenx, consisting in the generation of an IgG able to bind FcRn with a high affinity at acidic but also at neutral pH, making FcRn unavailable for the recycling of endogenous IgG. They provided the first in vitro and in vivo evidence of endogenous IgG depletion by FcRn blockade due to their lysosomal accumulation and destruction [75]. This discovery has opened the way to a new field of therapeutics, notably in autoantibody-mediated autoimmune diseases, by analogy with the use of plasma exchange and high doses of intravenous immunoglobulins (IVIg) [76,77].

### Enhancing mAb Affinity to FcRn at Acidic and Neutral pH to Degrade Endogenous IgG

The ABDEG technology allows increasing affinity of IgG for FcRn at acidic and neutral pH due to the M252Y/S254T/T256E-H433K/N434F (MST-HN) mutations located in the FcRn binding site. The long-lasting binding of an IgG with MST-HN mutations blocks the site for other IgG binding leading to endogenous IgG degradation with the expectation to degrade pathogenic autoantibodies in vivo [72].

Based on these results, a human IgG1 Fc fragment with MST-HN mutations has been developed. Called efgartigimod (G1e17), it reduced IgG levels up to 50% in healthy volunteers after a single 50 mg/kg administration without affecting the homeostasis of albumin [78]. Its mechanism of action as mentioned earlier is similar to that of IVIg in some diseases but without the latter’s side effects and with the advantage of not requiring the use of large amounts of human plasma donors, preventing the risk of shortages that have occurred repeatedly in recent years. Efgartigimod has been first approved by the FDA in 2021 for the treatment of myasthenia gravis [79] and is currently in a phase III study for the treatment of immune thrombocytopenia (ITP) [80], in phase II for the treatment of pemphigus vulgaris and foliaceus [81] and for that of chronic inflammatory demyelinating polyneuropathy [82].

As discussed above, “FcRn-blocking” agents lead to a decrease in IgG autoantibodies but also in any circulating IgGs, including those protecting the patients from infections. In view of this potential risk, the idea of selective depletion of particular antibodies has emerged, giving rise to Seldegs for selective degradation of antigen-specific antibodies. Seldegs are antigens or fragments of antigens linked to the human IgG1 Fc portion with the MST-HN mutations. In their princeps study, Devanaboyina et al. notably generated a Seldeg based on myelin oligodendrocyte glycoprotein (MOG). After injection of the MOG-Seldeg in an optic neuromyelopathy mouse model, a rapid depletion of circulating anti-MOG autoantibodies was observed without affecting the overall IgG level while improving the disease [83,84].

To mimic further IVIg, Fc fragments with enhanced FcRn and FcγR bindings are also being tested to target more specifically ICs. ICs containing autoantibodies can interact with FcγRs and play an important role in IgG-dependent pathophysiology such as cell cytotoxicity observed in autoimmune hemolytic anemia, immune thrombocytopenia (ITP), rheumatoid arthritis and systemic lupus erythematosus [85,86,87,88]. Monnet et al. have recently developed an Fc fragment with enhanced FcRn and FcγRIIIA binding affinities. The recombinant Fc LFBD192 with Y296W/K334N/P352S/A378V/V397M/N434Y mutations was compared in vitro to other relevant strategies such as IVIg or Fc carrying the MST-HN mutation with promising results. However, to our knowledge, there are currently no therapeutic trials involving LFBD192 [89]. The main localizations of mutations affecting IgG–FcRn interaction and their effects on mAbs are summarized in Figure 2.

## 4. Anti-FcRn Antibodies to Block IgG Binding on FcRn Leading to IgG Degradation including IgG Autoantibodies

In the same way as therapeutics relying on Fc engineering to enhance affinity for FcRn at both pH levels to degrade autoantibodies, other strategies ranging from peptides to full-length antibodies targeting FcRn directly have been developed. They all target the FcRn binding site without interfering with albumin uptake [90,91].

### 4.1. Full IgG Format

Anti-FcRn IgGs take up the lion’s share among mAbs in development for the treatment of autoimmune diseases [92]. They all have a silenced Fc preventing them from interacting with FcγR. There are currently two IgG4s, rozanolixizumab with an S241P mutation to stabilize the hinge [93] and orilanolimab [94], and two IgG1s, nipocalimab, a deglycosylated IgG1 [95], and batoclimab with L234A L235A mutations [96]. These mAbs have no effect on IgA, IgM and IgE concentrations, and no effect on FcRn expression, at least for orilanolimab [93,95,97]. Pharmacodynamic studies have shown the efficacy of all these drugs on the level of endogenous IgG in vivo [93,95,97]. Moreover, orilanolimab can decrease FcRn-mediated innate cytokine production by human peripheral blood leukocytes ex vivo and reduce circulating ICs within 5-6 days in vivo, and rozanolixizumab showed no cytokine induction ex vivo [93,97].

These drugs are in advanced stages of clinical trials for the treatment of myasthenia gravis [98,99,100]. Rozanolixizumab and batoclimab showed encouraging results in phase II trials in myasthenia gravis, reducing endogenous IgG levels ranging from 57% to 68% with a reduction in disease activity scores [99,101]. This result has been confirmed in a recent phase III study for rozanolixizumab [102]. Batoclimab is also being tested in other diseases such as Graves’ ophthalmopathy, idiopathic thrombocytopenic purpura, autoimmune hemolytic anemia and neuromyelitis optica [96]. Nipocalimab has also shown promising ex vivo results for the treatment of fetal anemia, by decreasing maternal IgG transfer to the fetus [103]; it is already being clinically evaluated for this disease (UNITY trial, NCT03755128) [104].

### 4.2. Other Formats

Other different FcRn inhibitors including small peptides, affibodies or Fc multimers have been designed. Small peptides such as SYN1436 (26 amino acids) for example were shown to decrease IgG levels up to 80% in cynomolgus monkeys by blocking protein interactions between IgG and FcRn [105,106]. Affibodies are small molecules engineered to bind to a target with high affinity, imitating mAbs. Z_FcRn_ is an anti-FcRn affibody used alone or fused to an albumin binding domain that showed a 40% reduction in endogenous IgG levels in mice after 5 days [107]. The only FcRn-targeting molecule tested in humans is ABY-039, developed by Alexion Pharmaceuticals Inc., which is a bivalent antibody-mimetic molecule of 19 kDa, able to bind to FcRn only at pH 6.0 [108]. Although promising, the phase I study was recently stopped [109]. Fc multimers are designed to inhibit IC-mediated FcγR or complement activation, FcRn being a side target, and have recently been reviewed by Zuercher et al. [92]. Two of them, a pan Fc receptor interacting molecule (PRIM) and CSL777, bind to FcRn but to our knowledge, there is no current clinical development [92].

## 5. Perspectives

As discussed throughout this review, FcRn is an attractive focus of interest that led to the generation of a diverse set of new therapeutic antibodies and related molecules. Its IgG recycling properties have been mainly exploited through mAb engineering or selection to increase their half-life or to decrease the level of IgG autoantibodies.

It is remarkable that such developments have been made possible without considering the cellular level of FcRn expression. Indeed, in the past ten years, we and others have described a link between FcRn expression, tumor cell growth and prognosis in cancer patients, whereby a low level of FcRn expression is related to a high rate of cell proliferation and bad prognosis in patients with lung and colon cancer [110,111,112]. A link between albumin consumption by tumor cells and low FcRn expression has also been described in cancer cell proliferation [110]. As such, it would seem that FcRn has yet to reveal all its secrets.

## Figures and Tables

**Figure 1 ijms-23-09604-f001:**
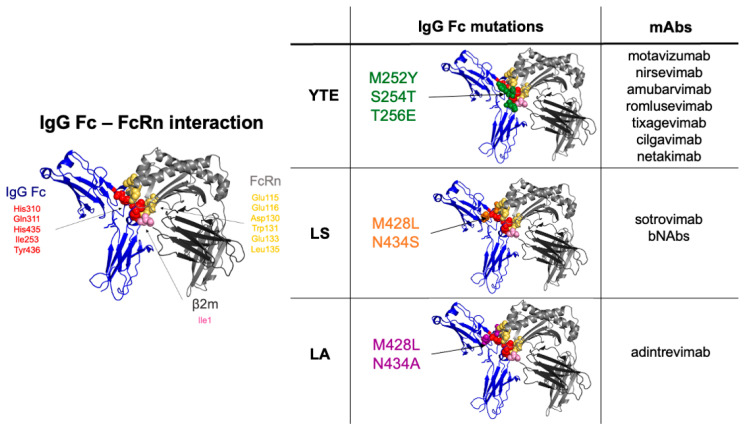
Three-dimensional view of IgG Fc–FcRn interaction and IgG Fc mutations in mAbs. IgG Fc, FcRn α-chain and β2-microglobulin are shown in dark blue, grey, and dark grey, respectively. IgG Fc–FcRn interaction is shown in red (IgG Fc), yellow (FcRn) and pink (β2-m) spheres. IgG Fc mutations in mAbs are shown as green (YTE), orange (LS) and purple (LA). The figures were made using PyMOL Molecular Graphics System Version 2.5.3 Copyright^©^ (Schrödinger) and the crystal structure data of IgG–Fc in complex with human FcRn (4N0U). *mAbs: monoclonal antibody*.

**Figure 2 ijms-23-09604-f002:**
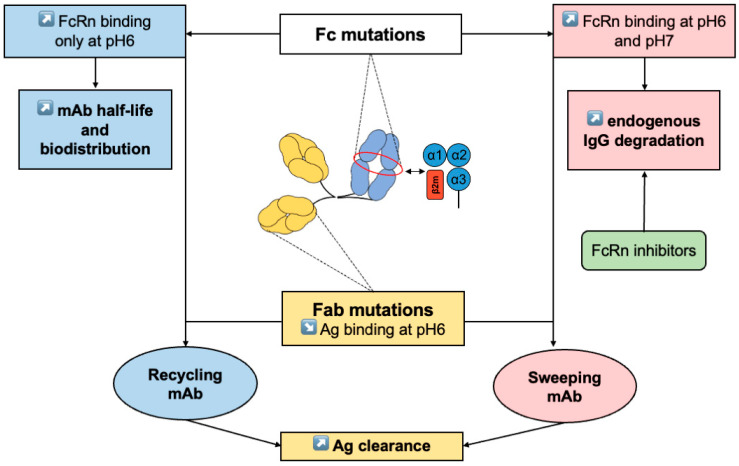
mAb mutations in Fc or Fab IgG. In Fc IgG, mutations can increase binding to FcRn at pH6 only (blue) or at pH6 and pH7 (pink) to increase mAb half-life and biodistribution or endogenous IgG degradation, respectively. FcRn inhibitors (green) can also increase endogenous IgG degradation. In Fab IgG (yellow), mutations can decrease IgG binding at pH6 and can be coupled to Fc IgG mutations (recycling or sweeping mAb) to increase Ag clearance. *mAb: monoclonal antibody*.

## Data Availability

Not applicable.

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
