# Peer review of "Monoclonal Antibody Engineering and Design to Modulate FcRn Activities: A Comprehensive Review"

_ijms, 2022, doi:10.3390/ijms23179604_

Round 1
Reviewer 1 Report
Abstract is a little bit short. Authors should add more information.
Not only increased risk of infection, but also several other adverse events associated with IVIG are reported. Authors should describe that point with mechanism of the adverse events.
Author Response
We would like to thank reviewer#1 for his relevant comments, please find below our responses.
Abstract is a little bit short. Authors should add more information.
- Response:
Abstract have been more detailed as requested
Not only increased risk of infection, but also several other adverse events associated with IVIG are reported. Authors should describe that point with mechanism of the adverse events
- Response:
Adverse events after IVIG administration are mainly represented by thrombosis (venous or arterial) leading to myocardial infarction and strokes due to plasma hyperviscosity. Another complication is acute kidney failure due to osmotic nephrosis. The occurrence of these 2 complications beeing reduced by hydration and fractionated administration. The last complication in our knowledge is hemolytic anemia due to the presence of allo hemagglutinins in the pooled IVIG.
We haven’t detailed these complications, because in our review, the main comparison is done with efgartigimod which does not present any of these adverse events according to the data published (Safety, efficacy, and tolerability of efgartigimod in patients with generalised myasthenia gravis (ADAPT): a multicentre, randomised, placebo-controlled, phase 3 trial. Howard, James FDe Bleecker, Jan L. et al.The Lancet Neurology, Volume 20, Issue 7, 526 – 536).
Please let us know if we have misunderstood the question and if our answer is not appropriate.
Reviewer 2 Report
The review is devoted to the modulation of monoclonal antibody interaction with FcRn receptor at acidic and neutral pH via specific mutations in the antibody sequence. This topic is highly significant as such engineered antibodies are widely used to treat various diseases. The review is very interesting, clearly written, quite detailed and I really delighted to read it. In my opinion, only minor corrections are required.
Specifically, very important and conceptual Figure 2 could be improved. The using of different colors and different shapes looks confusing. The meaning of arrow signs and “+” (above "sweeping mAbs") it is not clear at first glance. They could be replaced with words or denoted in figure's legend. Also, the phrase “IG degradation” suggests that it is the mutated antibodies that are being strenuously degraded, whereas in the text increased degradation of all the other antibodies is claimed.
Typos:
Line 304: “though” instead of “through”
Author Response
The review is devoted to the modulation of monoclonal antibody interaction with FcRn receptor at acidic and neutral pH via specific mutations in the antibody sequence. This topic is highly significant as such engineered antibodies are widely used to treat various diseases. The review is very interesting, clearly written, quite detailed and I really delighted to read it. In my opinion, only minor corrections are required.
Specifically, very important and conceptual Figure 2 could be improved. The using of different colors and different shapes looks confusing. The meaning of arrow signs and “+” (above "sweeping mAbs") it is not clear at first glance. They could be replaced with words or denoted in figure's legend. Also, the phrase “IG degradation” suggests that it is the mutated antibodies that are being strenuously degraded, whereas in the text increased degradation of all the other antibodies is claimed.
- Response:
We want to thank reviewer 2 for his kind and relevant comments. As requested, figure 2 has been modified in a way to make it more understandable. We have harmonized the colors with, in blue the Fc modification affecting the binding at acidic pH, in pink the Fc modifications affecting the binding at acidic and neutral pH and in yellow the fab modifications. The shapes have been simplified.
Typos: Line 304: “though” instead of “through”
Response:
The typo has been corrected.
Reviewer 3 Report
In this manuscript by Yanis Ramdani, et al., entitled " Monoclonal antibody engineering and design to modulate FcRn activities: a comprehensive review" summarized modification of antibodies to alter binding to FcRn. I thought the content was well organized and well written. To improve this manuscript, I suggest several modifications, which are listed below.
1. The residue number appears on line 68, please add the numbering rule (Kabat numbering?).
2. The authors describe the modification of tocilizumab CDRs in line 160, but the original paper names it PH2, so for clarity, please refer to it as, for example, “They engineered the tocilizumab CDRs and named it PH2”.
3. Classifications such as G1e16 appear in lines 101 and 221, but they are not common and confusing, so please remove them. If you need to include it, please do not indicate it in the text, but include it as a table summarizing the antibodies and classifications that appear in this paper.
4. In line 198, bi-specific is bispecific.
5. In line 189, 204, 215, please unify the notation of ABDEG technology.
6. In both Figures 1 and 2, the red wavy line, which appears to be derived from spell check, remains. Please fix them.
Author Response
In this manuscript by Yanis Ramdani, et al., entitled " Monoclonal antibody engineering and design to modulate FcRn activities: a comprehensive review" summarized modification of antibodies to alter binding to FcRn. I thought the content was well organized and well written. To improve this manuscript, I suggest several modifications, which are listed below.
- The residue number appears on line 68, please add the numbering rule (Kabat numbering?).
- The authors describe the modification of tocilizumab CDRs in line 160, but the original paper names it PH2, so for clarity, please refer to it as, for example, “They engineered the tocilizumab CDRs and named it PH2”.
- Classifications such as G1e16 appear in lines 101 and 221, but they are not common and confusing, so please remove them. If you need to include it, please do not indicate it in the text, but include it as a table summarizing the antibodies and classifications that appear in this paper.
- In line 198, bi-specific is bispecific.
- In line 189, 204, 215, please unify the notation of ABDEG technology.
- In both Figures 1 and 2, the red wavy line, which appears to be derived from spell check, remains. Please fix them.
- Response:
We want to thank the reviewer 3 for his relevant comments. All the modifications requested have been applied to the manuscript.